# Loss of functional BAP1 augments sensitivity to TRAIL in cancer cells

Krishna Kalyan Kolluri[1†], Constantine Alifrangis[2†], Neelam Kumar[1†], Yuki Ishii[1†], Stacey Price[2], Magali Michaut[3], Steven Williams[2], Syd Barthorpe[2], Howard Lightfoot[2], Sara Busacca[4], Annabel Sharkey[4], Zhenqiang Yuan[1], Elizabeth K Sage[1], Sabarinath Vallath[1], John Le Quesne[4], David A Tice[5], Doraid Alrifai[1], Sylvia von Karstedt[6], Antonella Montinaro[6], Naomi Guppy[7], David A Waller[8], Apostolos Nakas[8], Robert Good[9], Alan Holmes[9], Henning Walczak[6], Dean A Fennell[4], Mathew Garnett[2], Francesco Iorio[10], Lodewyk Wessels[3], Ultan McDermott[2]*, Samuel M Janes[1]*

[1]Lungs for Living Research Centre, UCL Respiratory, University College London, London, United Kingdom; [2]Wellcome Trust Sanger Institute, Cambridge, United Kingdom; [3]The Netherlands Cancer Institute, Amsterdam, Netherlands; [4]CRUK Leicester Centre, Department of Cancer studies, University of Leicester, Leicester, United Kingdom; [5]Oncology Research, MedImmune, Inc., Gaithersburg, United States; [6]Centre for Cell Death, Cancer and Inflammation, UCL Cancer Institute, University College London, London, United Kingdom; [7]UCL Advanced Diagnostics, University College London, London, United Kingdom; [8]Department of Thoracic Surgery, Glenfield Hospital, University Hospitals of Leicester, Leicester, United Kingdom; [9]UCL School of Pharmacy, University College London, London, United Kingdom; [10]European Molecular Biology Laboratory, European Bioinformatics Institute, Cambridge, United Kingdom

*For correspondence:
um1@sanger.ac.uk (UMD);
s.janes@ucl.ac.uk (SMJ)

†These authors contributed equally to this work

**Abstract** Malignant mesothelioma (MM) is poorly responsive to systemic cytotoxic chemotherapy and invariably fatal. Here we describe a screen of 94 drugs in 15 exome-sequenced MM lines and the discovery of a subset defined by loss of function of the nuclear deubiquitinase BRCA associated protein-1 (BAP1) that demonstrate heightened sensitivity to TRAIL (tumour necrosis factor-related apoptosis-inducing ligand). This association is observed across human early passage MM cultures, mouse xenografts and human tumour explants. We demonstrate that BAP1 deubiquitinase activity and its association with ASXL1 to form the Polycomb repressive deubiquitinase complex (PR-DUB) impacts TRAIL sensitivity implicating transcriptional modulation as an underlying mechanism. Death receptor agonists are well-tolerated anti-cancer agents demonstrating limited therapeutic benefit in trials without a targeting biomarker. We identify *BAP1* loss-of-function mutations, which are frequent in MM, as a potential genomic stratification tool for TRAIL sensitivity with immediate and actionable therapeutic implications.
DOI: https://doi.org/10.7554/eLife.30224.001

## Introduction

Amongst the most significant therapeutic breakthroughs in cancer has been the discovery of drug-sensitising genomic alterations. Drugs such as the tyrosine kinase inhibitors (TKIs) developed against the *BCR-ABL* fusion product in chronic myeloid leukaemia (CML) and the receptor products of *HER2* mutations in breast cancer have transformed the prognosis of these cancers (*Druker et al., 2006*). Malignant mesothelioma (MM) currently has no biomarker-driven therapies in routine clinical use.

**eLife digest** Two patients with the same disease who receive the same treatment may respond in different ways. This variation often arises from differences in each patient's genetic code. Genes encode proteins, and proteins are the targets of most medical drugs and thus determine the patient's response to treatment.

A major advance in the 21st century is that doctors recognise that patients can respond differently to the same treatment and now try to predict which patients will respond best to which drug – an approach known as personalised medicine. Cancer treatment has been at the forefront of personalised medicine because mutations in different genes underlie each different cancer. By analysing which mutations are present in a cancer, doctors can thus predict which drug (or combination of drugs) will be most effective. This approach has been used successfully in several cancers, including breast and lung cancer, leading to fewer patients being exposed to ineffective treatments and their associated side effects and costs.

Mesothelioma is a cancer of the lining of the lung that is associated with exposure to the mineral asbestos. Current treatment options for mesothelioma are unfortunately limited and not very effective. No personalised treatments are currently in use and new treatment approaches are desperately needed.

Kolluri, Alifrangis, Kumar, Ishii et al. set out to determine if any of the mutations commonly seen in mesothelioma affected how the cancer would respond to 94 anticancer drugs that are either in use or in development. In the laboratory, mesothelioma cells that have mutations in the gene that codes for a protein known as BRCA associated protein-1 (or BAP1 for short) were killed much more effectively by a drug known as TNF-related apoptosis-inducing ligand (TRAIL). The same link was seen in experiments with tumours of mesothelioma cells that had been transplanted into mice, and for fragments of mesothelioma tumours taken from patients. When Kolluri et al. studied why these tumours might be killed more effectively with TRAIL, they found that mutations in the gene for BAP1 result in a change in the levels of proteins that transmit the signal from the receptors targeted by the TRAIL drug.

These findings may one day result in a new approach to treating patients with mesothelioma. But first, the next step would be to conduct a clinical trial of TRAIL in patients with mesothelioma and assess if those with tumours that have mutations in the gene for BAP1 do indeed respond better. If this proves to be the case, this would result in a new personalised treatment option for patients that suffer from this disease.

DOI: https://doi.org/10.7554/eLife.30224.002

The mainstay of medical therapy for all patients remains systemic cytotoxic chemotherapy that offers only limited survival benefit in unselected populations; as such the disease remains invariably fatal (*Vogelzang et al., 2003*). A plethora of genomic studies in MM has identified recurrent mutations in several genes considered to be tumour drivers. *CDKN2A, NF2, BAP1* and *TP53* are the most frequently mutated (*Guo et al., 2015*; *Bueno et al., 2016*) and there has been increased focus on these genes and their associated signaling pathways as potential therapeutic targets (*LaFave et al., 2015*).

We aimed to determine if the mutational status of these tumour driver genes could predict response to a range of existing anti-cancer compounds with a view to identifying genomic biomarkers for responsive subsets of MM. We have previously reported on the ability of such unbiased high-throughput chemical screens in cancer cell lines to identify drug-sensitising mutations in other cancer types (*Garnett et al., 2012*). To this end, we conducted a high-throughput chemical screen of molecularly characterised MM cell lines seeking associations between MM driver gene mutations and compound response. This strategy led to the discovery of a subset of MM cell lines defined by loss-of-function (LOF) mutations in BRCA associated protein-1 (*BAP1*) that demonstrated heightened sensitivity to the death receptor agonist recombinant tumour necrosis factor (TNF)-related apoptosis-inducing ligand (rTRAIL). We validated this finding using *in vitro*, *in vivo* and *ex vivo* models supporting the use of *BAP1* as a genomic biomarker to identify TRAIL-sensitive MM tumours and a novel stratified approach to treat MM.

rTRAIL and other death receptor agonists selectively induce apoptosis in cancer cells and have long held promise as anti-cancer agents owing to their broad clinical utility and minimal off-target effects (*Wiley et al., 1995*; *Pitti et al., 1996*; *Ashkenazi et al., 1999*). Despite this, successful pre-clinical studies have not translated to clinical efficacy in trials of unselected populations (*Herbst et al., 2010*; *Wainberg et al., 2013*; *Soria et al., 2010*; *Lemke et al., 2014a*); there have been no trials to date in MM. However, within these trials some patients showed signs of therapeutic benefit and differential sensitivity within cell lines is well known. Retrospective biomarker identification has led to the stratified use of other anti-cancer therapies that initially failed in unselected trials such as activating *EGFR* mutations and EGFR TKIs (*Lynch et al., 2004*). We propose that BAP1 could potentially act as such a biomarker for the death receptor agonists. BAP1 is a nuclear deubiquitinase and forms multi-protein complexes that regulate the transcription of genes involved in key cellular functions including cell cycle regulation and DNA repair (*Ismail et al., 2014*; *Machida et al., 2009*). We investigated which BAP1 protein-binding partners, and thus which regulatory complexes, mediate TRAIL sensitivity identifying the BAP1-ASXL1 complex, the Polycomb repressive deubiquitinase (PR-DUB), as key. We further found that loss of BAP1 function modulates mRNA and protein expression of components of the extrinsic apoptotic pathway.

## Results

### A chemical screen uncovers genetic modifiers of drug response in mesothelioma

A 6 day viability screen using 94 drugs including small molecule inhibitors and cytotoxic chemotherapeutics (*Supplementary file 1*) was performed on 15 MM cell lines (*Supplementary file 2*) that had been characterised using whole-exome sequencing, copy number analysis and gene expression arrays. We generated 1425 single agent activity data profiles across the 15 cell lines (*Figure 1A* and *Supplementary file 3*). To detect novel markers of drug sensitivity, we sought statistical associations between drug response and the mutational status of the cell lines based on five genes identified as candidate drivers of tumourigenesis in MM (*Guo et al., 2015*) (*Figure 1—figure supplement 1*). There were 24 significant associations (false discovery rate (FDR) < 0.2) between single agent response and the presence of a genomic alteration. The most statistically significant sensitising association seen was between *BAP1* LOF mutations (mt *BAP1*) and treatment with recombinant TRAIL (rTRAIL; FDR = 0.18, effect size −0.48) (*Figure 1B,C* and *Supplementary file 4*). No significant effect on cell viability was observed in *BAP1* wild-type (wt *BAP1*) lines when treated with rTRAIL. We subsequently confirmed this association in a larger panel of MM cell lines (*Figure 1D* and *Supplementary file 5*). Strikingly, 6 of the 8 cell lines (75%) harbouring a *BAP1* LOF mutation were sensitive or partially sensitive to a dose range of rTRAIL while 7 of the 9 cell lines (78%) harbouring wild-type *BAP1* were resistant. *BAP1* LOF mutations correlated with a loss of BAP1 protein expression in the majority of cell lines (*Figure 1E*). No sensitising association with *BAP1* was observed for pemetrexed or cisplatin, which are current first line agents for the treatment of MM (*Figure 1—figure supplement 2A and B*). A marginal trend towards increased sensitivity in *BAP1* mutant MM lines in response to treatment with the agonistic FAS receptor antibody CH11 and a TNF-α/IAP inhibitor combination was observed. However, this was not as pronounced as that observed with rTRAIL or the multivalent death receptor five superagonist MEDI3039 (*Figure 1—figure supplement 2C,D and E*). Thus, while the significant sensitising association observed in the screen appears most specific to death receptor agonists, the trend observed with other TNF superfamily agonists indicates the BAP1-rTRAIL association to be mediated by an underlying mechanism common to this family such as the cytoplasmic extrinsic apoptotic machinery.

### The association of loss of BAP1 function with TRAIL sensitivity extends to other tumour types

To determine if knockdown of BAP1 in wild-type MM cells led to TRAIL sensitivity, we silenced *BAP1* expression in four wt *BAP1* MM cell lines using a lentiviral shRNA construct. Knockdown of *BAP1* resulted in increased cell death following rTRAIL treatment compared with empty vector (EV) control shRNA and the parental cell line in all four MM cell lines (*Figure 2A* and *Figure 2—figure supplement 1B and C* ). Loss of BAP1 expression has also been identified in several other tumour types

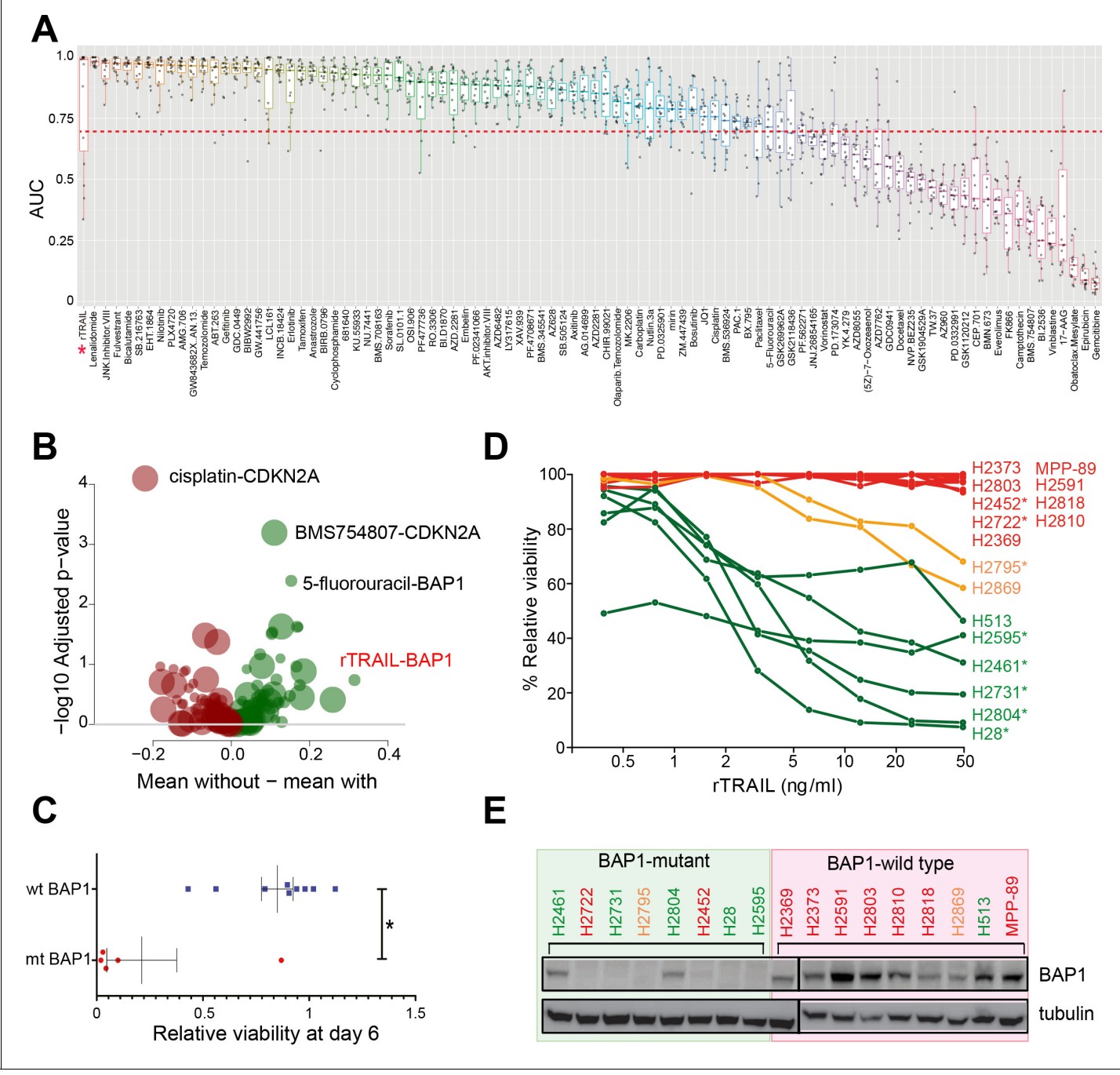

**Figure 1.** A chemical screen in mesothelioma cell lines identifies a BAP1-mutant population sensitised to the death receptor ligand rTRAIL. (A) Area under the curve (AUC) values for 15 malignant mesothelioma (MM) cells treated for 6 days with 94 compounds. Each dot indicates the AUC value for an individual cell line treated. AUC <0.7 is indicated by the red dotted line — only those compounds with ≥2 cell lines below this value were analysed for statistically significant associations with gene mutations. The AUC values for rTRAIL are indicated by the red asterisk. (B) A Welch t-test was used to test for significant pharmacogenomics interactions between the 94 single agents in the screen and the presence of driver mutations in any of 5 MM cancer genes. Each volcano plot circle corresponds to a significant gene–drug interaction whose position on the x-axis indicates the corresponding effect size. Both half-axes are positive; the right side (green circles) indicates the effect sizes of sensitivity associations, whereas the left side (red circles) corresponds with the effect sizes of resistance associations. The position on the y-axis indicates the statistical significance of the identified interaction. The size of a given circle is proportional to the number of samples in which the selected functional event involved in the corresponding interaction occurs. Specific examples of associations are indicated where the effect size is large (rTRAIL and *BAP1* mutations) or highly significant (cisplatin and *CDKN2A* mutations). (C) Cell viability between wild-type *BAP1* (wt BAP1) (n = 10) and mutant *BAP1* (mt BAP1) (n = 5) MM lines following 6 days of treatment with rTRAIL (t-test; *p=0.015). (D) Cell viability data for 17 MM lines treated for 6 days with a concentration range of rTRAIL (0.4–50 ng/ml).
*Figure 1 continued on next page*

Figure 1 continued

MM lines are coloured according to their sensitivity pattern (green = sensitive (**S**); orange = partially sensitive (PS); red = resistant (**R**)). *Indicates cell lines harbouring BAP1 mutations. (**E**) Immunoblot of BAP1 protein expression in *BAP1*-mutant versus *BAP1*-wild-type MM cell lines. Sensitivity to rTRAIL treatment is indicated as font colour: green (**S**); orange (PS); red (**R**).
DOI: https://doi.org/10.7554/eLife.30224.003
The following figure supplements are available for figure 1:
**Figure supplement 1.** Mutation status of 5 candidate tumour driver genes in the 15 MM lines used in the combinatorial chemical inhibitor screen.
DOI: https://doi.org/10.7554/eLife.30224.004
**Figure supplement 2.** BAP1 and the response to alternative apoptotic stimuli in MM cells.
DOI: https://doi.org/10.7554/eLife.30224.005

including uveal melanoma (47%) (*Harbour et al., 2010*), clear cell renal carcinoma (CCRC) (14%) (*Peña-Llopis et al., 2012*) and cholangiocarcinoma (7%) (*Fujimoto et al., 2015*). Notably, knockdown of BAP1 in two CCRC lines resulted in increased sensitivity to rTRAIL in addition to the MDAMB-231 breast cancer line (*Figure 2B* and *Figure 2—figure supplements 2* and *3*). We also analysed a panel of 1001 cancer cell lines submitted for whole exome and copy number analysis as part of the COSMIC cell lines project (*Forbes et al., 2015*) and identified nine additional non-mesothelioma cell lines harbouring truncating mutations in *BAP1* (http://cancer.sanger.ac.uk/cancergenome/projects/cell_lines/). These include CCRC, bladder and breast cancer lines. Treatment of cancer cell lines harbouring nonsense mutations in *BAP1* with rTRAIL resulted in markedly reduced cell viability compared with cancer cell lines harbouring missense mutations (*Figure 2—figure supplement 4*).

## BAP1 modulates TRAIL sensitivity through PR-DUB activity

BAP1 is a nuclear deubiquitinase that forms multi-protein complexes with transcription factors to regulate gene transcription (*Jensen et al., 1998*; *Ventii et al., 2008*). To elucidate the mechanism by which BAP1 modulates sensitivity to TRAIL we generated expression vectors containing wild-type or mutant forms of *BAP1*, each with an inactive functional site or protein-binding domain. These included C91A (mutation in the deubiquitination catalytic site) (*Jensen et al., 1998*; *Ventii et al., 2008*), ΔHBM (deletion of the HCF-1-binding site) (*Misaghi et al., 2009*), T493A (mutation in the FOXK2-binding site) (*Ji et al., 2014*), ΔASXL (deletion of the ASXL1/2 protein-binding site) (*Daou et al., 2015*) and ΔCTD (deletion of the C-terminal domain containing the nuclear localisation signal) (*Ventii et al., 2008*). H226 MM cells, which harbour a homozygous deletion of *BAP1* and demonstrate complete loss of BAP1 expression, were transduced with a GFP (vector control), a wild-type *BAP1* expression vector or one of these five mutant *BAP1* expression vectors. rTRAIL sensitivity of the parental *BAP1*-null H226 MM line was significantly diminished following expression of wild-type *BAP1* and each of the mutant constructs except those with an inactive deubiquitinating or ASXL protein-binding site (*Figure 2C*), implicating the function of these sites in BAP1-induced TRAIL resistance. These effects were replicated using MEDI3039 (*Figure 2D*). Transduction of two further BAP1-mutant rTRAIL-sensitive cell lines, H28 and H2804, with wild-type BAP1 also induced resistance to rTRAIL while sensitivity was maintained with transduction of the deubiquitinase mutant (*Figure 2—figure supplement 5*).

The BAP1 deubiquitinase and ASXL-binding sites are key to the function of the PR-DUB, an epigenetic transcriptional regulatory complex composed of BAP1 and ASXL1. Deubiquitination of the main substrate of the PR-DUB, H2AK119Ub, alters chromatin architecture to modulate gene transcription (*Scheuermann et al., 2010*). This led us to hypothesise that PR-DUB, rather than exclusively BAP1, function might underlie rTRAIL sensitivity. Consistent with this shRNA silencing of *ASXL1*, but not *ASXL2*, induced sensitivity to MEDI3039 and rTRAIL in the *BAP1/ASXL1/ASXL2*-wild-type MM line MPP-89 (*Figure 2E* and *Figure 2—figure supplement 6*). Furthermore, H2AK119Ub expression was unaltered in the rTRAIL-sensitive H226 cells transduced with mutant constructs that disrupt PR-DUB activity, while the rTRAIL-resistant H226 cells transduced with a wild-type BAP1 construct exhibited lower H2AK119Ub levels (*Figure 2—figure supplement 7*). Thus, as the PR-DUB complex is implicated in transcriptional regulation, differential modulation of specific transcriptional programmes by BAP1 may determine rTRAIL sensitivity. We therefore compared differential gene expression data from *BAP1*-null H226 cells transduced with the C91A *BAP1* mutant or with wild-type

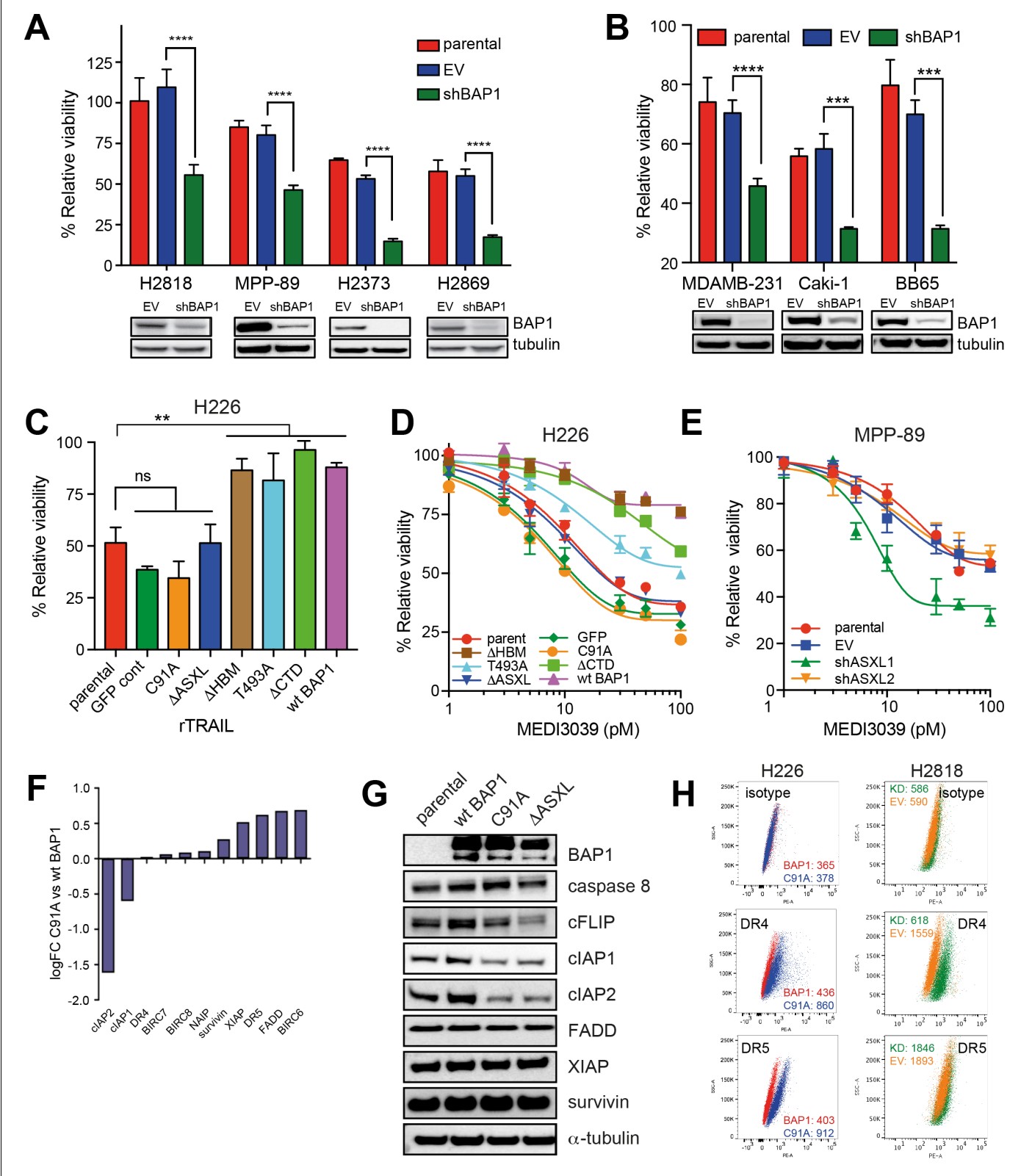

**Figure 2.** BAP1-induced TRAIL resistance extends to other cancer subtypes and is dependent upon functional deubiquitinase and ASXL-binding sites. (A) *BAP1*-wild-type H2818, MPP-89, H2373 and H2869 MM lines were transduced with BAP1 (shBAP1) or empty vector (EV) shRNA. Immunoblot confirmed BAP1 knockdown in the BAP1 shRNA-transduced cells. Parental and transduced cells were treated with rTRAIL (1000 ng/ml) and cell viability assessed after 72 hr by MTT assay (t-test; ****p<0.0001). (B) The *BAP1*-wild-type breast cancer line MDAMB-231 and the renal cell carcinoma (RCC) lines

*Figure 2 continued on next page*

*Figure 2 continued*

Caki-1 and BB65 were transduced with BAP1 (shBAP1) or empty vector (EV) shRNA. Immunoblot confirmed BAP1 knockdown in the BAP1 shRNA transduced cells. Parental and transduced cells were treated with rTRAIL (1000 ng/ml) and cell viability assessed after 72 hr by MTT assay (t-test; ****p<0.0001). (C) The rTRAIL-sensitive H226 MM line, which harbours a homozygous deletion of *BAP1*, was transduced with either a GFP control, wild-type *BAP1* or a mutant *BAP1* containing an inactive functional domain: C91A — inactivating mutation of deubiquitinase catalytic site; ΔHBM — deletion of HCF-1-binding motif; T493A — inactivating mutation of FOXK2-binding site; ΔASXL — deletion of ASXL1/2 protein-binding site; ΔCTD — deletion of C-terminal domain containing nuclear localisation signal. These transduced lines were treated with 50 ng/ml rTRAIL and cell death assessed with XTT assay (one-way ANOVA; **p<0.01). (D) The parental and transduced H226 MM lines were treated with a concentration range (1–100 pM) of the small molecule death receptor agonist MEDI3039 and cell viability assessed with XTT assay. (E) The *BAP1*-wild-type MPP-89 MM line was transduced with ASXL1 (shASXL1), ASXL2 (shASXL2) or empty vector (EV) shRNA. qPCR confirmed a decrease in ASXL1 and ASXL2 mRNA expression in the ASXL1 shRNA and ASXL2 shRNA-transduced cells, respectively (*Figure 2—figure supplement 6*). Parental and transduced cells were treated with a concentration range (1–100 pM) of MEDI3039 and cell viability assessed with XTT assay. (F) Differential gene expression of apoptosis regulator genes in the catalytically inactive BAP1-mutant (C91A) relative to the wild-type BAP1-transduced (wt BAP1) H226 cells. (G) Immunoblot of apoptosis regulator proteins in the catalytically inactive BAP1-mutant (C91A), inactive ASXL1/2-binding site BAP1-mutant (ΔASXL) or wild-type BAP1-transduced (wt BAP1) H226 cells. (H) Flow cytometry analysis of death receptor 4 (DR4) and 5 (DR5) cell surface expression in H226 cells transduced with the catalytically inactive *BAP1*-mutant (C91A) or wild-type *BAP1* (wt BAP1) and of *BAP1*-wild-type H2818 MM cells transduced with BAP1 (KD) or empty vector (EV) shRNA. The values represent the median fluorescence intensity (MFI).

DOI: https://doi.org/10.7554/eLife.30224.006

The following figure supplements are available for figure 2:

**Figure supplement 1.** shRNA knockdown of *BAP1* increases sensitivity to rTRAIL in MM cells.
DOI: https://doi.org/10.7554/eLife.30224.007

**Figure supplement 2.** shRNA knockdown of *BAP1* increases sensitivity to DR agonists in breast cancer cells.
DOI: https://doi.org/10.7554/eLife.30224.008

**Figure supplement 3.** shRNA knockdown of *BAP1* increases sensitivity to DR agonists in clear cell renal carcinoma cells.
DOI: https://doi.org/10.7554/eLife.30224.009

**Figure supplement 4.** Cell viability of non-mesothelioma *BAP1*-mutant cell lines following rTRAIL treatment.
DOI: https://doi.org/10.7554/eLife.30224.010

**Figure supplement 5.** Overexpression of wild-type *BAP1* induces resistance to rTRAIL in *BAP1* mutant MM cells.
DOI: https://doi.org/10.7554/eLife.30224.011

**Figure supplement 6.** shRNA knockdown of *ASXL1* increases sensitivity of MM cells to rTRAIL.
DOI: https://doi.org/10.7554/eLife.30224.012

**Figure supplement 7.** Ubiquitinated histone 2A at K119 (H2AK119Ub) expression and BAP1 function.
DOI: https://doi.org/10.7554/eLife.30224.013

**Figure supplement 8.** Differential gene expression data from H226 MM cells expressing C91A-mutant (mt BAP1) or wild-type *BAP1* (wt BAP1).
DOI: https://doi.org/10.7554/eLife.30224.014

**Figure supplement 9.** Signalling pathway impact analysis of gene expression data from H226 MM cells expressing C91A-mutant (mt BAP1) or wild-type *BAP1* (wt BAP1).
DOI: https://doi.org/10.7554/eLife.30224.015

*BAP1* and carried out a signalling pathway impact analysis (SPIA) ((*Figure 2—figure supplements 8 and 9* [SPIA_H226 C91A mutant vs WT]) (http://www.genome.jp/dbget-bin/www_bget?path: map04210). Among those pathways significantly altered when comparing wild-type versus C91A *BAP1* (FDR < 0.2) was that of apoptosis. In particular, there was altered mRNA expression of components of the extrinsic death pathway (*Figure 2F* and *Supplementary file 6*). This manifested as an imbalance in levels of pro- and anti-apoptotic mRNA expression with, for example, significantly decreased levels of the anti-apoptotic cIAP1/2 (p=2.32E-10) and increased levels of the pro-apoptotic death receptor 5 (p=7.79E-10) in the rTRAIL sensitive C91A *BAP1*-transduced cells relative to the rTRAIL resistant *BAP1*-wild-type transduced cells. Immunoblot analysis confirmed reduced protein expression of cIAP1/2 and c-FLIP in both C91A and ΔASXL *BAP1*-transduced cells relative to *BAP1*-wild-type transduced cells (*Figure 2G*). Flow cytometry analysis confirmed reduced DR4 and DR5 expression in C91A BAP1 transduced relative to *BAP1*-wild-type-transduced cells. Knockdown of *BAP1* in the *BAP1* wild-type H2818 line resulted in a significant increase in DR4 expression only (*Figure 2H*).

## BAP1 loss-of-function sensitises human early passage mesothelial cell lines, human tumour explants and mouse mesothelioma xenograft models to rTRAIL

To support the clinical relevance of our finding we extended our assays to two further models derived from primary tumour tissue. 25 human early passage MM lines from the UK Mesobank (*Rintoul et al., 2016*) were assessed for BAP1 expression by immunohistochemistry, a technique known to correlate strongly with *BAP1* LOF mutations in the absence of strong nuclear staining (*Nasu et al., 2015*). When treated with rTRAIL, those without strong nuclear staining were significantly more sensitive than those with strong nuclear staining (p=0.0067). Of the 12 lines that did not express nuclear BAP1 9 were sensitive, 2 partially sensitive and only one resistant to rTRAIL (*Table 1*, *Figure 3A* and *Figure 3—figure supplement 1*). Remarkably, rTRAIL treatment of tumour explants derived from three patients with MM also revealed increased levels of apoptosis (as measured by poly (ADP-ribose) polymerase (PARP) cleavage) in explants with low BAP1 expression compared with those with high BAP1 expression (*Figure 3B and C*, *Figure 3—figure supplement 2*).

To test the *in vivo* efficacy of TRAIL in inducing apoptosis in *BAP1*-mutant MM cells, we transduced the H226 *BAP1*-wild-type and the H226 C91A *BAP1*-mutant cell lines with luciferase and injected equal numbers of wild-type and mutant cells into the opposite flanks of mice (*Figure 3—figure supplement 3A*). On day 14 after injection the mice were divided into two groups and injected intraperitoneally with rTRAIL or vehicle for 6 days per week until day 40. At sacrifice rTRAIL-treated

**Table 1.** BAP1 immunoblot status, nuclear BAP1 staining and rTRAIL sensitivity (50 ng/ml) of the 25 human early passage MM cultures.

| Sample name | Western blot | Nuclear BAP1-IHC | Sensitivity |
|---|---|---|---|
| 7T | − | − | Sensitive |
| 8T | − | − | Sensitive |
| 45 | − | − | Sensitive |
| 19 | − | − | Sensitive |
| 14T | − | − | Sensitive |
| 12 | − | − | Sensitive |
| 23T | − | − | Sensitive |
| 40 | − | − | Sensitive |
| 36 | Low Expression | − | Sensitive |
| 26 | + | + | Sensitive |
| 12T | + | + | Sensitive |
| 3T | + | + | Sensitive |
| 52 | − | − | Partially Sensitive |
| 2 | − | − | Partially Sensitive |
| 30 | Low Expression | + | Partially Sensitive |
| 15 | Low Expression | + | Partially Sensitive |
| 35 | + | + | Partially Sensitive |
| 24 | + | + | Partially Sensitive |
| 43 | − | − | Resistant |
| 34 | + | + | Resistant |
| 50T | + | + | Resistant |
| 33T | + | + | Resistant |
| 18 | + | + | Resistant |
| 53T | + | + | Resistant |
| 38 | + | + | Resistant |

DOI: https://doi.org/10.7554/eLife.30224.016

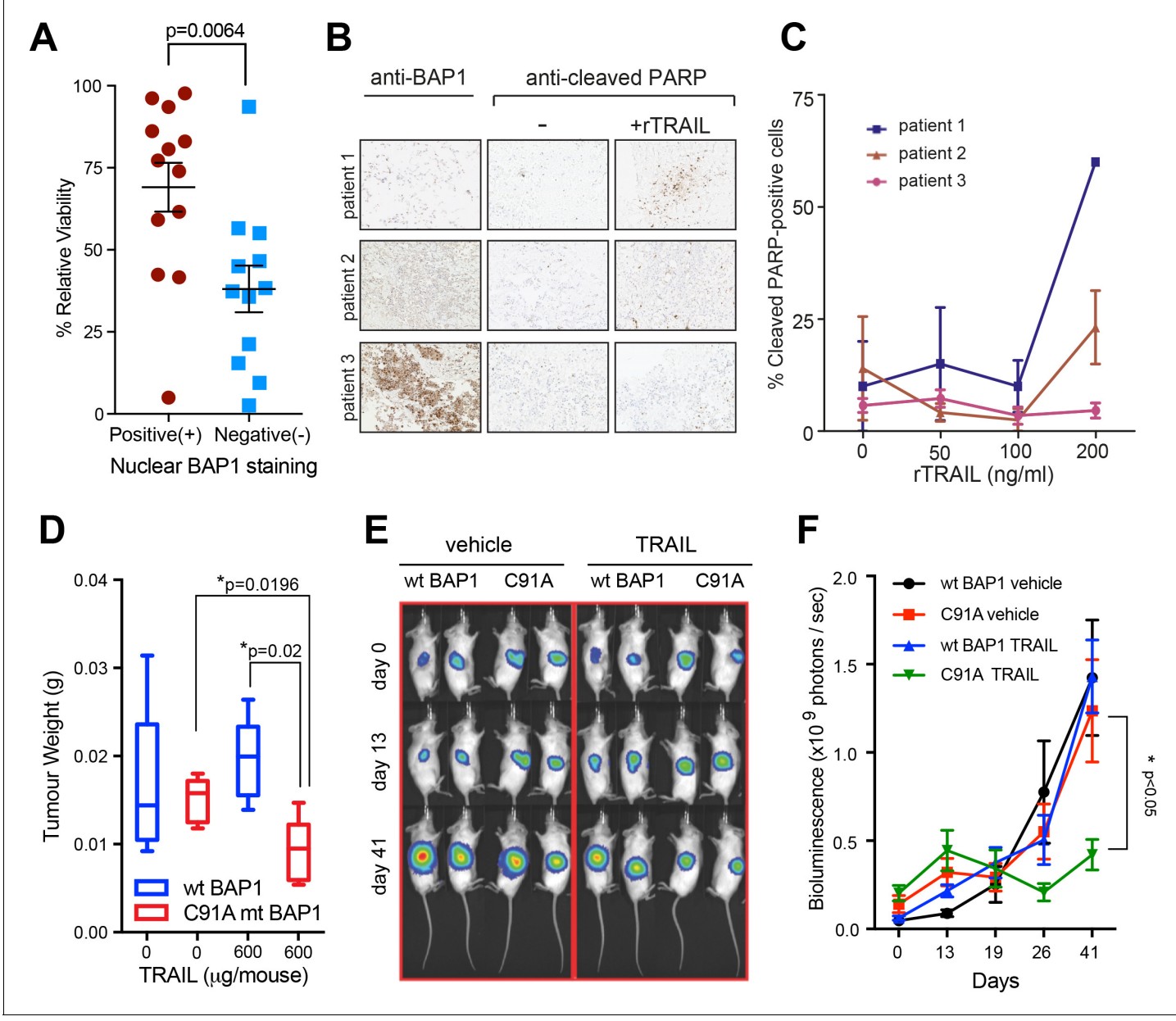

**Figure 3.** Loss of functional BAP1 leads to TRAIL sensitivity in early passage mesothelioma cell lines, human tumour explants and mouse xenograft models. (**A**) Mean cell viability effect between human early passage MM cell lines (positive nuclear BAP1 staining; n = 13 and negative nuclear BAP1 staining; n = 12) as assessed by immunohistochemistry following 3 days of treatment with rTRAIL (50 ng/ml) (t-test, p=0.0067). (**B**) Immunohistochemical images of tumour explants derived from three MM patients treated with either vehicle or rTRAIL for 24 hr. Explants were stained with anti-BAP1 and anti-cleaved PARP (marker for apoptosis) antibodies. (**C**) The percentage of cleaved PARP-positive cells in tumour explants derived from three patients and treated with either vehicle or 0, 50, 100 and 200 ng/ml of rTRAIL for 24 hr was scored based on the percentage of cells with nuclear cleaved PARP-positive staining. (**D**) Weights of tumour xenografts derived from *BAP1*-wild-type (wt BAP1) versus catalytically inactive BAP1-mutant (C91A mt BAP1) transduced MM cells following treatment with either vehicle or TRAIL (600 μg per mouse) at the time of sacrifice (day 42) (t-test). (**E**) Serial bioluminescence imaging of *BAP1*-wild-type (wt BAP1) and catalytically inactive *BAP1*-mutant (C91A) MM xenografts in mice treated with either vehicle or TRAIL. Mice were imaged on day 0 (after tumour inoculation), day 13 (before TRAIL administration) and day 41 (time of sacrifice). The intensity of luminescence is denoted by colour: red - high luciferase signal (high tumour burden) and blue - low luciferase signal (low tumour burden). (**F**) A time-course of bioluminescence scores in *BAP1*-wild-type (wt BAP1) versus catalytically inactive *BAP1*-mutant (C91A) MM tumour xenografts. Bioluminescence was measured on days 0, 13, 19, 26 and 41, 15 min after injecting the mice with 0.2 ml luciferin intraperitoneally. (two way ANOVA).

DOI: https://doi.org/10.7554/eLife.30224.017

The following figure supplements are available for figure 3:

*Figure 3 continued on next page*

*Figure 3 continued*

**Figure supplement 1.** BAP1 expression in early passage MM cultures.
DOI: https://doi.org/10.7554/eLife.30224.018
**Figure supplement 2.** *Ex vivo* experimental protocol.
DOI: https://doi.org/10.7554/eLife.30224.019
**Figure supplement 3.** *In vivo* experimental protocol.
DOI: https://doi.org/10.7554/eLife.30224.020

---

BAP1-mutant tumours weighed significantly less than rTRAIL-treated *BAP1*-wild-type tumours (p=0.020) and vehicle-treated *BAP1*-mutant tumours (p=0.019) (*Figure 3D* and *Figure 3—figure supplement 3B*). *BAP1*-wild-type tumours showed no response to rTRAIL compared with vehicle. The growth rate of rTRAIL-treated *BAP1*-mutant tumours was also significantly suppressed compared with rTRAIL-treated *BAP1*-wild-type and vehicle-treated tumours (p<0.05) as assessed by longitudinal bioluminescence intensity (*Figure 3E and F*).

## Discussion

Malignant mesothelioma remains a devastating disease with limited systemic treatment options (*Vogelzang et al., 2003*). Biomarker-driven therapies have significantly improved the prognosis for subsets of patients within other cancer types however this strategy has yet to impact MM. Our data support the use of loss of function of BAP1 as a genomic stratification tool to identify rTRAIL-sensitive MM tumours, an approach that may extend to other cancer subtypes. We propose the underlying mechanism involves the transcriptional regulation of expression of components of the apoptotic pathway by the PR-DUB. Our finding has potentially significant and immediately actionable clinical implications for both MM treatment and for the death receptor agonist field.

BAP1 has emerged as a key driver of tumorigenesis in MM (*Bueno et al., 2016*). As such, there has been increased focus on this nuclear deubiquitinase and its associated pathways (*LaFave et al., 2015*). While next-generation sequencing reveals MM *BAP1* mutation rates in the order of 20–30% (*Guo et al., 2015*; *Bueno et al., 2016*; *Bott et al., 2011*), immunohistochemical analysis has identified loss of BAP1 function in up to 67% of MM tumours (*Nasu et al., 2015*) opening our biomarker-driven approach to a significant proportion of MM patients. BAP1 immunohistochemistry accurately identifies loss of BAP1 function as a consequence of genetic and non-genetic mechanisms (*Nasu et al., 2015*) and is already in clinical use as a diagnostic tool; hence the clinical tools for our proposed approach are validated and ready. Our data indicate the BAP1-TRAIL association extends beyond MM to other tumours with loss of BAP1 function. Chromosomal deletions and somatic inactivating mutations have been identified at high frequency in uveal melanoma (*Harbour et al., 2010*), clear cell renal carcinoma (*Peña-Llopis et al., 2012*) and cholangiocarcinoma (*Fujimoto et al., 2015*), increasing the potential clinical impact of our discovery. Although loss of BAP1 function is seen at far lower rates in breast carcinoma (1%) (*Stephens et al., 2012*) and non-small cell lung carcinoma (1%) (*Owen et al., 2017*), the high incidence of these cancers translates to a large cohort of patients.

Focus on death receptor agonists as anti-cancer agents has generated two decades of preclinical studies and the development of numerous clinically tested compounds, all of which have demonstrated limited therapeutic efficacy at phase I/II trials (*Herbst et al., 2010*; *von Pawel et al., 2014*; *Paz-Ares et al., 2013*; *Forero-Torres et al., 2013*). Strategies to overcome this have included the development of increasingly potent death receptor agonists and combination therapies to address resistance factors within the apoptosis pathway (*Holland, 2013*; *Lemke et al., 2014b*). As differential sensitivity has been observed in trials, it has been accepted that identification of a biomarker predicting the therapeutic outcome is of paramount importance (*Ashkenazi, 2015*; *von Karstedt et al., 2017*). There have been previous attempts to identify predictive biomarkers largely focused on molecular expression panels (*Passante et al., 2013*). Ours is the first unbiased approach to address how the genetic make-up of tumours predicts response to rTRAIL treatment. The identification of *BAP1* as a potential genomic biomarker has the potential to reignite the death receptor agonist field of research into which significant investment has already been made. The value of retrospective analysis of clinical trials based on the genomic landscape has clearly been

demonstrated in the past (*Lynch et al., 2004*) and we wait with interest whether this will be performed on archived tumour tissue, in the context of *BAP1* status, from previous trials. Notably there have been no trials of any death receptor agonists in MM or indeed any cancer with a high proportion of *BAP1* mutations. We suspect a significantly higher proportion of responders would have been identified in such trials.

Our findings also have implications for death receptor agonists as a therapy for *BAP1*-wild-type tumours as delineation of the underlying mechanism would offer a novel avenue by which to sensitise these tumours. Our mechanistic data implicate transcriptional regulation by the PR-DUB as key to the capacity of BAP1 to modulate death receptor agonist sensitivity. BAP1 is a master genetic regulator and is known to influence the transcription of thousands of genes as supported by our and others' gene expression data (*Dey et al., 2012*). While we highlight the extrinsic apoptotic pathway and proteins as being significantly altered by *BAP1* status, identifying a single factor to explain BAP1-induced TRAIL resistance is extremely challenging. Of more direct clinical significance is our finding that loss of function of either component of the PR-DUB, BAP1 or ASXL1, results in an increase in death receptor agonist sensitivity. *ASXL1* mutations have an important role in the pathogenesis of myeloid neoplasms primarily consisting of nonsense, missense and frameshift mutations resulting in a truncated ASXL1 protein that retains the BAP1-binding domain (*Boultwood et al., 2010*). It has yet to be clarified if this truncated protein possesses dominant-negative or gain-of-function properties in the context of PR-DUB activity (*Balasubramani et al., 2015*). In the case of the former, *ASXL1* could potentially predict death receptor agonist sensitivity in myeloid neoplasms. Further research is needed in these malignancies to determine this.

Confirmation of the clinical value of BAP1 as a targeting biomarker for death receptor agonists in early phase clinical trials of mesothelioma is the first priority. The clinical tools for this approach are already validated and established facilitating the translation of our discovery into a desperately needed new therapy for this fatal thoracic cancer.

# Materials and methods

**Key resources table**

| Reagent type (species) or resource | Designation | Source or reference | Identifiers | Additional information |
|---|---|---|---|---|
| gene | | | | |
| *BRCA associated protein-1* (human) | *BAP1* | Entrez Gene NCBI | Gene ID: 8314 | |
| *Additional sex combs like 1* (human) | *ASXL1* | Entrez Gene NCBI | Gene ID: 171023 | |
| strain, strain background | | | | |
| NOD.CB17-Prkdc^scid/NcrCrl | NOD SCID mice | Charles River Laboratories, UK | RRID:IMSR_CRL:394 | |
| cell line | | | | |
| Early passage mesothelioma cell cultures | 7T, 8T, 45, 19, 14T, 23T, 40, 36, 26, 12T, 3T, 52, 2, 30, 15, 35, 24, 43, 34, 50T, 33T, 18, 53T, 38 | MesobanK, Mesothelioma UK | | www.mesobank.com Mesothelioma Tissue Bank, Papworth Hospital NHS Trust, UK |
| NCI-H2373 | H2373 | Wellcome Trust Sanger Institute, UK | RRID:CVCL_A533 | |
| NCI-H2803 | H2803 | Wellcome Trust Sanger Institute, UK | RRID:CVCL_U997 | |
| NCI-H2452 | H2452 | Wellcome Trust Sanger Institute, UK | RRID:CVCL_1553 | |
| NCI-H2722 | H2722 | Wellcome Trust Sanger Institute, UK | RRID:CVCL_U994 | |

*Continued on next page*

Continued

| Reagent type (species) or resource | Designation | Source or reference | Identifiers | Additional information |
|---|---|---|---|---|
| NCI-H2369 | H2369 | Wellcome Trust Sanger Institute, UK | RRID:CVCL_A532 | |
| NCI-H2795 | H2795 | Wellcome Trust Sanger Institute, UK | RRID:CVCL_U996 | |
| NCI-H2869 | H2869 | Wellcome Trust Sanger Institute, UK | RRID:CVCL_V001 | |
| NCI-H2591 | H2591 | Wellcome Trust Sanger Institute, UK | RRID:CVCL_A543 | |
| MPP 89 | MPP-89 | Wellcome Trust Sanger Institute, UK | RRID:CVCL_1427 | |
| NCI-H2810 | H2810 | Wellcome Trust Sanger Institute, UK | RRID:CVCL_U999 | |
| NCI-H2818 | H2818 | Wellcome Trust Sanger Institute, UK | RRID:CVCL_V000 | |
| NCI-H513 | H513 | Wellcome Trust Sanger Institute, UK | RRID:CVCL_A570 | |
| NCI-H2595 | H2595 | Wellcome Trust Sanger Institute, UK | RRID:CVCL_A545 | |
| NCI-H2461 | H2461 | Wellcome Trust Sanger Institute, UK | RRID:CVCL_A536 | |
| NCI-H2731 | H2731 | Wellcome Trust Sanger Institute, UK | RRID:CVCL_U995 | |
| NCI-H2804 | H2804 | Wellcome Trust Sanger Institute, UK | RRID:CVCL_U998 | |
| NCI-H28 | H28 | Wellcome Trust Sanger Institute, UK | RRID:CVCL_1555 | |
| NCI-H226 | H226 | Szlosarek lab, Barts Cancer Institute, UK | RRID:CVCL_1544 | |
| MDA-MB-231 | MDAMB-231 | Wellcome Trust Sanger Institute, UK | RRID:CVCL_0062 | |
| Caki-1 | Caki-1 | Wellcome Trust Sanger Institute, UK | RRID:CVCL_0234 | |
| BB65 | BB65 | Wellcome Trust Sanger Institute, UK | RRID:CVCL_1078 | |
| antibody | | | | |
| BAP1 (C-4) mouse mAb | anti-BAP1 | Santa Cruz Biotechnology, Santa Cruz, CA | Cat# sc-28383 RRID:AB_626723 | 1:500 in milk; 1:50 for flow cytometry |
| Caspase-8 (1C12) mouse mAb | anti-caspase 8 | Cell Signaling Technology, Danvers, MA | Cat# 9746 RRID:AB_2275120 | 1:1000 in BSA |
| FLIP (7F10) mouse mAb | anti c-FLIP | Enzo Life Sciences, Farmingdale, NY | Cat# ALX-804-961-0100 RRID:AB_2713915 | 1:1000 in milk |
| c-IAP1 (D5G9) rabbit mAb | anti-cIAP1 | Cell Signaling Technology, Danvers, MA | Cat# 7065S RRID:AB_10890862 | 1:1000 in BSA |
| c-IAP2 (58C7) rabbit mAb | anti-cIAP2 | Cell Signaling Technology, Danvers, MA | Cat# 3130S RRID:AB_10693298 | 1:1000 in BSA |
| FADD rabbit pAb | anti-FADD | Cell Signaling Technology, Danvers MA | Cat# 2782 RRID:AB_2100484 | 1:1000 in BSA |
| XIAP (3B6) rabbit mAb | anti-XIAP | Cell Signaling Technology, Danvers, MA | Cat# 2045 RRID:AB_2214866 | 1:1000 in milk |
| survivin rabbit pAb | anti-survivin | Cell Signaling Technology, Danvers, MA | Cat# 2803 RRID:AB_490807 | 1:1000 in BSA |

*Continued*

| Reagent type (species) or resource | Designation | Source or reference | Identifiers | Additional information |
|---|---|---|---|---|
| α-Tubulin (11H10) Rabbit mAb | anti-α-tubulin | Cell Signaling Technology, Danvers, MA | #2125 | 1:2000 in milk |
| Ubiquityl-Histone H2A (Lys119) (D27C4) XP Rabbit mAb | anti-H2AK119Ub | Cell Signaling Technology, Danvers, MA | Cat# 8240P RRID:AB_10891618 | 1:2000 in BSA |
| Histone H2A (D6O3A) Rabbit mAb | anti-H2A | Cell Signaling Technology, Danvers, MA | Cat# 12349 RRID:AB_2687875 | 1:1000 in BSA |
| Anti-mouse IgG, HRP-linked antibody | anti-mouse HRP | Cell Signaling Technology, Danvers, MA | Cat# 7076 RRID:AB_330924 | 1:2000 in milk |
| Anti-rabbit IgG, HRP-linked antibody | anti-rabbit HRP | Cell Signaling Technology, Danvers, MA | Cat# 7074 RRID:AB_2099233 | 1:2000 in milk |
| Donkey anti-Mouse IgG (H + L) Highly Cross-Adsorbed Secondary Antibody, AlexaFluor 488 | AlexaFluor 488-conjugated anti-mouse antibody | Thermo Fisher Scientific, UK | Cat# A-21202 RRID:AB_141607 | 1:200 for flow cytometry |
| Annexin V, AlexaFluor 647 conjugate | Annexin V AlexaFluor 647-conjugated antibody | Thermo Fisher Scientific, UK | Cat# A23204 RRID:AB_2341149 | 1:100 for flow cytometry |
| PE anti-human CD261 (DR4, TRAIL-R1) antibody | PE-conjugated antibody to DR4 | Biolegend, UK | Cat# 307205 RRID:AB_314669 | 1:100 for flow cytometry |
| PE anti-human CD262 (DR5, TRAIL-R2) antibody | PE-conjugated antibody to DR5 | Biolegend, UK | Cat# 307405 RRID:AB_314677 | 1:100 for flow cytometry |
| PE Mouse IgG1, κ Isotype Ctrl Antibody | PE isotype control antibody | Biolegend, UK | Cat# 400112 | 1:100 for flow cytometry |
| Goat anti-Rabbit IgG (H + L) Secondary Antibody, AlexaFluor 488 conjugate | AlexaFluor 488-conjugated anti-rabbit secondary antibody | Thermo Fisher Scientific, UK | Cat# R37116 RRID:AB_2556544 | 1:200 for flow cytometry |
| Anti-Cleaved PARP1 (E51) mAb | cleaved PARP primary antibody; anti-cleaved PARP | Abcam, UK | Cat# ab32064 RRID:AB_777102 | (1:6000) for immunohistochemistry |
| recombinant DNA reagent | | | | |
| BAP1 (NM_004656) Human cDNA Clone | pCMV6-AC BAP1 plasmid | Origene, Rockville, MD | Cat# SC117256 | |
| pHIV-Luc-ZsGreen | ZS-green luciferase plasmid, pHIV-Luc-ZsGreen | Bryan Welm Lab, University of Utah, Addgene, Logan, UT | Cat# 39196 | |
| pCMVR8.74 | pCMV-dR8.74 | Thrasher lab, UCL, Addgene, UK | Cat# 22036 | |
| pMD2.G | pMD2.G | Thrasher lab, UCL, Addgene, UK | Cat# 12259 | |
| sequence based reagent | | | | |
| BAP1 GIPZ Lentiviral shRNA | BAP1 shRNA | UCL RNAi Library (Dharmacon, Lafayett, CO) | V2LHS 41473 | |
| ASXL1 GIPZ Lentiviral shRNA | ASXL1 shRNA | UCL RNAi Library (Dharmacon, Lafayett, CO) | V2LHS 78829 | |
| ASXL2 GIPZ Lentiviral shRNA | ASXL2 shRNA | UCL RNAi Library (Dharmacon, Lafayette, CO) | V3LHS_313940 | |
| peptide, recombinant protein | | | | |
| Recombinant Human sTRAIL | rTRAIL | Peprotech, UK | Cat# 310–04 | |
| commercial assay or kit | | | | |

*Continued on next page*

*Continued*

| Reagent type (species) or resource | Designation | Source or reference | Identifiers | Additional information |
|---|---|---|---|---|
| Cell Proliferation Kit XTT | XTT reagent | Applichem, UK | A8088 | |
| Q5 Site-Directed Mutagenesis Kit | Site directed mutagenesis | New England Biolabs, Ipswich, MA | Cat# E0554 | |
| Rabbit specific HRP/DAB (ABC) Detection IHC Kit | rabbit-specific HRP/DAB (ABC) detection IHC kit | Abcam, UK | Cat# ab64261 | |
| chemical compound, drug | | | | |
| MEDI3039 | MEDI3039 | MedImmune, UK | | |
| software, algorithm | | | | |
| GraphPad Prism software | Graphpad Prism | GraphPad Software, CA, USA | | |
| CaVEMan algorithm | CaVEMan | https://github.com/cancerit/CaVEMan | | |
| Pindel algorithm | Pindel | https://github.com/genome/pindel | | |
| Predicting Integral Copy Numbers In Cancer algorithm | PICNIC | http://www.sanger.ac.uk/science/tools/picnic | | |
| FlowJo software | Flowjo | FlowJo LLC | | |
| Other | | | | |
| RIPA buffer | RIPA | Sigma-Aldrich, St. Louis, MO | Cat# R0278 | |
| Syto™ 60 red fluorescent nucleic acid stain | Syto 60 | Thermo Fisher Scientific, UK | Cat# S11342 | |
| Thiazolyl Blue Tetrazolium Bromide (MTT) | MTT reagent | Sigma-Aldrich, St. Louis, MO | Cat# M2128 | |
| jetPEI DNA transfection reagent | jetPEI | Source Bioscience, UK | Cat# 101–10 | |
| Polybrene | Polybrene | Sigma-Aldrich, St Louis, MO | Cat# 107689 | 8 µg/ml |
| Hoechst 33342 Solution (20 mM) | Hoechst 33342 | Thermo Fisher Scientific, UK | Cat# 62249 | |
| 4', 6-diamidino-2-phenylindole | DAPI | Sigma-Aldrich, St Louis, MO | Cat# D9542 | 200 µg/ml |

## Drug screens

### Drugs in the screen

Compounds were from academic collaborators or commercial vendors. Each compound, its therapeutically relevant target substrate and pathway and the minimum and maximum screening concentrations are listed in *Supplementary file 1*. Compounds were stored as 10 µM aliquots at −80°C and were subjected to a maximum of 5 freeze-thaw cycles. For the screen a fixed single 40 ng/ml concentration of rTRAIL was used, while each of the 94 agents was screened at a 5-point serial 4-fold dilution to give a 256-fold range from the lowest to highest concentration. The concentrations selected for each compound were based on *in vitro* data to cover the range of concentrations known to inhibit relevant kinase activity and cell viability.

### Genomic/transcriptomic characterization of mesothelioma cell lines

The genomic data is available in the COSMIC database (*Forbes et al., 2015*) (http://cancer.sanger.ac.uk/cancergenome/projects/cell_lines/).

### Substitution and insertion/deletion variant data

Exome sequencing was carried out using the Agilent SureSelectXT Human All Exon 50 Mb bait set giving an average 7 Gb of unique mapped reads per sample with an average of 85% of base pairs covered to >20 reads. Variants were identified by comparison to a reference single unmatched normal sample. Differences from the reference genome were identified using the CaVEMan and Pindel algorithms identifying substitution and small insertions/deletions respectively (https://github.com/

cancerit/CaVEMan; https://github.com/genome/pindel) (*Ye et al., 2009*). The resulting variants were then screened against approximately 8000 normal samples to remove sequencing artefacts and germline variants (428 in-house normal exomes, 6500 normal exomes (NHLBI GO Exome Sequencing Project, June 20th 2012 release), 1000 genomes project (29th March 2012 release) and variants in the dbSNP database that had an associated minor allele frequency.

## Copy number data

Genome-wide copy number data were obtained for the cell lines using the Affymetrix SNP6 microarray analysed using the 'PICNIC' algorithm, which segments the genome into integer value copy number segments (*Greenman et al., 2010*) (http://www.sanger.ac.uk/genetics/CGP/Software/PICNIC/). All genes were mapped onto this segmentation data to give a gene level copy number analysis. For genes to be classified as amplified the complete coding footprint of the gene had to map onto segment(s) present in eight or more copies. For genes to be classed as homozygously deleted a minimum of 1 bp of coding sequence had to be present within a segment of copy number '0'.

## Cell viability assay in compound screen

Cells were seeded in either 96-well or 384-well microplates in RPMI-1640 or DMEM/F12. The optimal cell number for each cell line was determined to ensure that each was in growth phase at the end of the assay (~70% confluency). Adherent cell lines in the screens were plated 1 day prior to treatment with each compound using liquid handling robotics and assayed after 6 days of treatment with either the single agent or in combination with rTRAIL. Cells were fixed in 4% formaldehyde for 30 min and then stained with 1 µM of the fluorescent nucleic acid stain Syto 60 (Thermo Fisher Scientific, UK) for 1 hr. Quantitation of fluorescent signal intensity was performed using a fluorescent plate reader at excitation and emission wavelengths of 630/695 nm. The sensitivity of each cell line to various concentrations of compound was calculated as the fraction of viable cells relative to DMSO-treated cells following a 6 day exposure. All screening plates were subjected to stringent quality control measures and a Z-factor score comparing negative and positive control wells was calculated across all screening plates (median = 0.70, upper quartile = 0.86, lower quartile = 0.47, n = 4857 plates).

## Calculation of AUC values from cell line viability data

We derived the area under the curve (AUC) parameter from the 6 day cell line viability data to identify cell lines that are sensitive to a specific compound, with decreasing AUC associated with increasing sensitivity. The AUCs were computed using a trapezoid integration below the five measured viability of the dose-response curve and scaled so that a constant viability of 1 gives AUC of 1.

## Statistical analysis of the effect of genetic features on drug response

We used 15 mesothelioma cell lines with molecular and drug response data: H2369, H2373, H2461, H2591, H2722, H2731, H2803, H2804, H2810, H2818, H2869, H513, MPP-89, NCI-H2452 and NCI-H28. We selected five genes for inclusion in the analysis (*BAP1*, *TAOK1*, *NF2*, *TP53* and *CDKN2A*). We defined groups of cell lines based on mutations and copy number alterations (homozygous deletions or amplifications) in these genes. This resulted in a set of input features of 4 genes altered in at least 2 of the cell lines (*CUL1*, *RDX* and *PIK3C2B* were not mutated and *TAOK1* was only mutated in 1 cell line). For the association of gene mutations with sensitivity to each compound we restricted the set of drugs to test to those with ≥2 cell lines with AUC <0.7. This resulted in 45 drugs being suitable for analysis (of the overall 94 drugs).

## Cell lines

All cell lines were sourced from the Wellcome Trust Sanger Institute except the H226 line that was a kind gift from Dr Peter Szlosarek, Barts Cancer Institute. All cell lines were authenticated by genotyping using Short Tandem Repeat (STR) and Sequenom profiling of a panel of 92 single nucleotide polymorphisms for each cell line to ensure non-synonymous cell lines were not used. As a cell line classified as mesothelioma, H513 (on the list of commonly misidentified cell lines) was included in the drug screen of 15 mesothelioma cell lines conducted. Use of this cell line however was not carried forward to further experiments in the paper. The 25 early passage MM cultures were purchased

from MesobanK (*Rintoul et al., 2016*). All cell lines and cultures were tested for mycoplasma contamination and confirmed to be negative.

## Cell culture

Cell lines were cultured in RPMI-1640 or Dulbecco's modified Eagle's medium and nutrient mix 12 medium (DMEM:F12) supplemented with 10% fetal bovine serum (FBS), penicillin/streptavidin and sodium pyruvate. Early passage human mesothelioma cultures were cultured in RPMI-1640 medium supplemented with 5% FBS, 25 mM HEPES, penicillin/streptavidin and sodium pyruvate. 293 T cells were cultured in Dulbecco's modified Eagle's medium (DMEM) supplemented with 10% fetal bovine serum (FBS) and 2 mM L-glutamine. All cells were maintained in a humidified environment at 37°C and 5% $CO_2$.

## Immunoblotting and antibodies

Cells were lysed in radioimmunoprecipitation assay (RIPA) buffer (Sigma-Aldrich, St. Louis, MO) with protease inhibitors (Complete-mini; Roche, Switzerland) on ice to extract protein. 20 µg of protein samples were separated by SDS–PAGE and transferred onto nitrocellulose membranes. Membranes were incubated with specific primary antibodies, washed, incubated with secondary antibodies and visualised using an ImageQuant LAS 4000 imaging system (GE Healthcare, Little Chalfont, NY). Antibodies used include BAP1 (Santa Cruz Biotechnology, Santa Cruz, CA) Cat# sc-28383, RRID:AB_626723), caspase 8 (Cell Signaling Technology, Danvers, MA) Cat# 9746, RRID:AB_2275120), c-FLIP (Enzo Life Sciences, Farmingdale, NY) Cat# ALX-804-961-0100 RRID:AB_2713915), cIAP1 (Cell Signaling Technology Cat# 7065S, RRID:AB_10890862), cIAP2 (Cell Signaling Technology Cat# 3130S, RRID:AB_10693298), FADD (Cell Signaling Technology Cat# 2782, RRID:AB_2100484), XIAP (Cell Signaling Technology Cat# 2045, RRID:AB_2214866), survivin (Cell Signaling Technology Cat# 2803, RRID:AB_490807), α-tubulin (Cell Signaling #2125), H2AK119Ub (Cell Signaling Technology Cat# 8240P, RRID:AB_10891618), H2A (Cell Signaling Technology Cat# 12349, RRID:AB_2687875), anti-mouse HRP (Cell Signaling Technology Cat# 7076, RRID:AB_330924) and anti-rabbit HRP (Cell Signaling Technology Cat# 7074, RRID:AB_2099233). To detect the ubiquitination status of the histones, the cells were lysed with TBS buffer containing 1% SDS, protease and phosphatase inhibitors. The cell extract was denatured by heating up at 95°C for 10 min and centrifuged at 13000 rpm for 10 min. The supernatant was collected and immunoblotted as described above.

## XTT/MTT cell viability assay

Cells were seeded in 96-well plates in 100 µl media per well at a density of 40,000 cells/ml 1 day prior to treatment with soluble recombinant TRAIL (rTRAIL; Peprotech, UK) or MEDI3039 (Medimmune, UK). XTT (Applichem, UK; A8088) or MTT (M-2128, Sigma-Aldrich) reagent was added on day 3. The absorbance was measured with a spectrophotometer at a wavelength of 490 nm or 560 nm for XTT or MTT respectively. Relative cell viability was calculated as a fraction of viable cells relative to untreated cells.

## Plasmids

Full-length *BAP1* cDNA was amplified by PCR from pCMV6-AC *BAP1* plasmid (Origene (Rockville, MD; SC117256) and cloned into the lentiviral plasmid pCCL-CMV-flT vector previously described (*Yuan et al., 2015*) in place of flT via BamHI and SalI sites, creating the *BAP1* vector designated pCCL-CMV-BAP1. Vectors expressing mutant *BAP1* constructs were generated by site-directed mutagenesis (New England Biolabs) of the pCCL-CMV-BAP1 vector. The primers used are listed below. All mutations were confirmed by sequencing.

 BAP1-F CGTGGATCCGCCACCATGAATAAGGGCTGGCTGGA
 BAP1-R GTCGGTCGACTCACTGGCGCTTGGCCTTGTA
 C91A-F ATACCCAACTCTGCTGCAACTCATGCCTTGCTG
 C91A-R CAGCTGGTGGGCAAAGAACATGTTATTCACAATATCATC
 HBM-F CGCTGCTGCCAAGTCCCCCATGCAGGAGGA
 HBM-R GCAGCGTCTAGAAAGGCCGGCAGCCGCT
 CTD-F CGTGGATCCGCCACCATGAATAAGGGCTGGCTGGA
 CTD-R GTCGTTCGAATCAGTCAGGCTTCCGCTGCTTGTGG

T493A-F GCAGACACGGCCTCTGAGATCGGCAGTGCT
T493A-R ACTCTCATTGCTGGGGGTGGGTGA
ASXL-F AACTACGATGAGTTCATCTGCACCT
ASXL-R CTGGTCATCAATCTTGAACTTCTTCCTC

The ZS-green luciferase plasmid, pHIV-Luc-ZsGreen (a gift from Bryan Welm, Addgene plasmid #39196) was used for generating ZS-Green luciferase-expressing lentivirus to transduce the H226 cells used in animal experiments.

## RNA interference

Short hairpin RNAs (shRNAs) were expressed as part of a mir30-based GIPZ lentiviral vector (Dharmacon, Lafayette, CO). The clones used in this study include BAP1 (V2LHS_41473), ASXL1 (V2LHS_78829), ASXL2 (V3LHS_313940) and the empty GIPZ control vector.

## Lentivirus production and cell transfection

Lentiviral vectors were produced by co-transfection of 293 T cells with construct plasmids together with the packaging plasmids pCMV-dR8.74 and pMD2.G (kind gifts from Dr Adrian Thrasher, UCL, Addgene plasmid #22036 and #12259) in the presence of a DNA transfection reagent jetPEI (Source Bioscience UK Ltd). Lentiviruses were concentrated by ultracentrifugation at 17,000 rpm (SW28 rotor, Optima LE80K Ultracentrifuge, Beckman Coulter, Brea, CA) for 2 hr at 4°C. To determine the titres of prepared lentiviruses 293 T cells were transduced with serial dilutions of viruses in the presence of 8 µg/ml Polybrene (Sigma-Aldrich) and BAP1 expression was assessed by flow cytometry. shRNA- and luciferase-expressing vectors were assessed by analysis of GFP expression. Cell lines were transduced in the presence of 8 µg/ml Polybrene at a range of MOIs and transduction efficacy was assessed by flow cytometry for BAP1 expression.

## Gene expression analyses

We pre-processed and normalised raw CEL files from Affymetrix Human Genome U219 array plate hybridisations with the Multi-Array Average (RMA) method (*Irizarry et al., 2003*). We discarded transcripts with low sample variance and consolidated duplicated genes by averaging their expression values across duplicates. The resulting data were subsequently normalised ($\mu = 0$, $\sigma = 1$) sample-wise and gene-median centred. Gene expression was averaged across three biological replicates of H226 transduced cells with either a C91A mutant or a wild-type *BAP1* construct. SPIA pathway analysis as described in Tarca *et al* (*Tarca et al., 2009*) was performed on those genes with an adjusted $p < 0.05$ and a fold change of >1.

## Flow cytometry

All flow cytometry analysis was performed on a LSR Fortessa analyser (Becton Dickinson, Franklin Lakes, NJ). For analysis of BAP1 expression cells were stained with primary antibody to BAP1 (Santa Cruz Biotechnology Cat# sc-28383, RRID:AB_626723; 1:50) and then with an AlexaFluor 488-conjugated anti-mouse antibody (Thermo Fisher Scientific Cat# A-21202, RRID:AB_141607; 1:200). For analysis of apoptosis and cell death all floating and adherent cells were harvested and stained with an Annexin V AlexaFluor 647-conjugated antibody (Thermo Fisher Scientific Cat# A23204, RRID:AB_2341149) and 4', 6-diamidino-2-phenylindole (DAPI; Sigma-Aldrich, 200 µg/ml). For analysis of DR4 and DR5 expression on cell surface cells were stained with PE-conjugated antibody (DR4 - BioLegend, San Diego, CA) Cat# 307205, RRID:AB_314669, DR5 - BioLegend Cat# 307405, RRID:AB_314677, Isotype control - Biolegend #400112; 1:100). FlowJo software was used to analyse all data.

## Immunofluorescence

H226 cells were seeded at $2.5 \times 10^3$ cells per well into 96-well Greiner micro-clear imaging plates in DMEM 10% FBS. After 48 hr, cells were fixed in 4% PFA for 10 min at room temperature and permeabilised in 0.3% NP-40 in PBS for 10 min. Cells were blocked in 1% BSA in 0.1% PBS tween for 1 hr at room temperature. Ubiquityl-histone H2A (Lys119) primary antibody (Cell Signaling, #8240) was incubated overnight at 4°C, before incubating for 1 hr at room temperature with Alexafluor 488-conjugated anti-rabbit secondary antibody. Nuclei were stained with Hoechst 33342 (Thermo Fisher Scientific Cat# 62249). Images were acquired (n = 3) with a BioTek Cytation3 Multimode reader.

Using a 10x objective 4 fields of view were acquired per well (n = 3) and the level of nuclear ubiquityl-histone H2A intensity was determined within the primary nuclear mask and normalised to total cell number.

## Immunohistochemical analysis of early passage cultures

BAP1 immunohistochemistry of human early passage cell lines was conducted on sections of cell pellets mounted on slides. Automated staining on a Leica Bond III staining platform was used. Slides were incubated with BAP1 primary antibody (Santa Cruz Biotechnology Cat# sc-28383, RRID:AB_626723; 1:150) for 15 min at room temperature. Epitope retrieval was completed using HIER using Leica Bond ER2 (high pH) for 30 min and a Leica Bond Polymer Refine with DAB chromogen detections system used.

## Mesothelioma patient explants

Appropriate ethical approval was obtained from the NHS Health Research Authority National Research Ethics Service to carry out this work (reference 14/LO/1527). Informed consent to conduct research on samples collected and to publish results was obtained from patients. The diagnosis of mesothelioma was confirmed histologically for all patients prior to consent and surgery. Patients underwent pleurectomy, following which primary pleural tissue was sectioned into fragments measuring approximately 2 $mm^3$. These tissue explants were cultured in 50% neurobasal and 50% DMEM:F12, supplemented with B27 (2%), EGF (20 ng/ml) and FGF (10 ng/ml). After 24 hr the explants were treated with rTRAIL (vehicle, 50 ng/ml, 100 ng/ml or 200 ng/ml) for a further 24 hr, following which explants were either fixed for PARP immunohistochemistry. The explants were fixed in 10% neutral-buffered formalin (NBF) for 24 hr and then transferred into 70% ethanol followed by paraffin embedding. Subsequently, 5 µm sections were used for immunohistochemistry, as previously described (Busacca et al., 2016).

### Immunohistochemistry of patient explants

Cleaved PARP primary antibody (Abcam Cat# ab32064, RRID:AB_777102) was used at a 1:6000 dilution and the rabbit-specific HRP/DAB (ABC) detection IHC kit (Abcam) was used for immunohistochemistry, according to the manufacturer's instructions. Sections were counterstained with haematoxylin and mounted using Vectamount permanent mounting media (Vector Labs, Peterborough, United Kingdom). Images were taken at 40x magnification on a Hamamatsu Nanozoomer Digital slide scanner. Cleaved PARP-positive cells were scored as the percentage of cells with nuclear staining.

## Animals

All animal studies were approved by the University College London Biological Services Ethical Review Committee and licensed under the UK Home Office regulations and the Evidence for the Operation of Animals (Scientific Procedures) Act 1986 (Home Office, London, UK). Mice were purchased from Charles River, kept in individually ventilated cages under specific pathogen-free conditions and had access to sterile irradiated food and autoclaved water ad libitum.

## Xenograft mouse models

12 8 week old NOD.CB17-Prkdcscid/NcrCrl (NOD SCID) mice (Charles River, UK; RRID:IMSR_CRL:394) were injected with 1 $\times$ $10^6$ H226 cells transduced with a plasmid containing wild-type BAP1 and luciferase on the right flank and with a plasmid containing a catalytically inactive BAP1-mutant (C91A) and luciferase on the left flank in a 1:1 mixture of Matrigel (Corning, Corning, NY) and medium. Tumour size was assessed by bioluminescence in vivo imaging system (IVIS, PerkinElmer, Waltham, MA) 15 min following intraperitoneal injection of 0.2 ml (2 mg) luciferin. Tumours were allowed to establish for 2 weeks prior to baseline assessment of size at day 13. Mice were then divided into two groups each of which received either 600 µg TRAIL or vehicle 6 days a week from day 14 until day 40. Bioluminescence was measured on days 0, 13, 19, 26 and 41. Mice were sacrificed on day 42 and tumours harvested for measurement. TRAIL used in the mouse experiment was made in Henning Walczak's laboratory as per the established protocol (Ganten et al., 2006).

## Statistical analysis

Statistical analysis was performed using GraphPad Prism (GraphPad Software, CA, USA). t-test was used to analyse differences between two groups whilst the analysis of variance (ANOVA) test with a Tukey post-hoc analysis was used to analyse differences between three groups. For multiple groups measured over multiple time points repeated measures ANOVA was used. All *in vitro* tests were performed in triplicate and all data are represented as mean values ± standard error of mean unless otherwise stated.

## Acknowledgements

The authors would like to thank Dr Kate Gowers and Dr Rob Hynds (UCL Respiratory) for proof reading the manuscript.

## Additional information

### Competing interests

David A Tice: Employed in MedImmune, Inc. Henning Walczak: Cofounder and shareholder of Apogenix AG. The other authors declare that no competing interests exist.

### Funding

| Funder | Grant reference number | Author |
|---|---|---|
| Wellcome | WT097452MA | Constantine Alifrangis |
| Wellcome Trust | 106555/Z/14/Z | Neelam Kumar |
| Cancer Research UK | A17341 | Henning Walczak |
| Cancer Research UK | | Ultan McDermott |
| Wellcome | WT107963AIA | Samuel M Janes |

The funders had no role in study design, data collection and interpretation, or the decision to submit the work for publication.

### Author contributions

Krishna Kalyan Kolluri, Constantine Alifrangis, Conceptualization, Data curation, Formal analysis, Validation, Investigation, Visualization, Methodology, Writing—original draft, Writing—review and editing; Neelam Kumar, Data curation, Investigation, Writing—original draft, Writing—review and editing; Yuki Ishii, Mathew Garnett, Data curation, Formal analysis, Validation, Investigation, Methodology; Stacey Price, Syd Barthorpe, Sabarinath Vallath, Data curation, Investigation; Magali Michaut, Howard Lightfoot, Data curation, Formal analysis, Visualization; Steven Williams, Antonella Montinaro, Data curation, Methodology; Sara Busacca, Sylvia von Karstedt, Data curation, Methodology, Writing—review and editing; Annabel Sharkey, Robert Good, Data curation; Zhenqiang Yuan, Formal analysis, Investigation, Methodology; Elizabeth K Sage, Data curation, Investigation, Methodology; John Le Quesne, Formal analysis, Validation, Methodology; David A Tice, Resources; Doraid Alrifai, Data curation, Validation; Naomi Guppy, Alan Holmes, Data curation, Formal analysis, Methodology; David A Waller, Apostolos Nakas, Provided surgical samples; Henning Walczak, Resources, Methodology; Dean A Fennell, Formal analysis, Visualization, Methodology; Francesco Iorio, Data curation, Formal analysis, Visualization, Methodology; Lodewyk Wessels, Data curation, Formal analysis, Investigation, Visualization, Methodology; Ultan McDermott, Samuel M Janes, Conceptualization, Resources, Data curation, Formal analysis, Supervision, Funding acquisition, Validation, Investigation, Visualization, Methodology, Writing—original draft, Project administration, Writing—review and editing

## Author ORCIDs

Krishna Kalyan Kolluri  http://orcid.org/0000-0003-3537-0231
Constantine Alifrangis  http://orcid.org/0000-0002-5876-6696
Neelam Kumar  http://orcid.org/0000-0002-4346-2112
Magali Michaut  http://orcid.org/0000-0003-2002-2277
Steven Williams  http://orcid.org/0000-0002-9382-4337
Elizabeth K Sage  http://orcid.org/0000-0002-9009-421X
Samuel M Janes  http://orcid.org/0000-0002-6634-5939

## Ethics

Human subjects: Appropriate ethical approval was obtained from the NHS Health Research Authority local National Research Ethics Committee Service to carry out this work (reference 14/LO/1527). Informed consent to conduct research on samples collected and to publish results was obtained from patients

Animal experimentation: All animal studies were approved by the University College London Biological Services Ethical Review Committee and licensed under the UK Home Office regulations and the Evidence for the Operation of Animals (Scientific Procedures) Act 1986 (Home Office, London, UK).

## Decision letter and Author response

Decision letter https://doi.org/10.7554/eLife.30224.031
Author response https://doi.org/10.7554/eLife.30224.032

# Additional files

## Supplementary files

• Supplementary file 1. List of 94 compounds used either as single agents or in combination with rTRAIL. Listed are the unique ID number, the compound name and target, the cellular process targeted and the minimum and maximum concentration (micromolar) of the 5-point concentration range used for each compound.
DOI: https://doi.org/10.7554/eLife.30224.021

• Supplementary file 2. Name and histological subtype (where known) of the 15 mesothelioma cell lines.
DOI: https://doi.org/10.7554/eLife.30224.022

• Supplementary file 3. 1425 area under the curve (AUC) viability scores for 94 experimental agents tested against 15 mesothelioma cell lines after 6 days of treatment.
DOI: https://doi.org/10.7554/eLife.30224.023

• Supplementary file 4. Results of Welch's two sample t-test from analysis of 45 single compounds that ≥2 cell lines demonstrated sensitivity to (AUC <0.7) and using the mutation status of eight genes implicated as drivers in mesothelioma in each cell line. A 6 day viability assay was used to determine cell line sensitivity. False discovery associations < 0.2 are highlighted as red font. Whether a mutation is associated with resistance or sensitivity to that compound is indicated by red or green in the 'effect' column, respectively.
DOI: https://doi.org/10.7554/eLife.30224.024

• Supplementary file 5. Description of *BAP1* mutations detected in 15 mesothelioma cell lines and the sensitivity of the cell lines to rTRAIL (as measured by a 6 day viability assay). The sensitivity of each cell line is indicated in the last column as sensitive (green), partially sensitive (orange) or resistant (red).
DOI: https://doi.org/10.7554/eLife.30224.025

• Supplementary file 6. Differential gene expression values of apoptotic genes in H226 mesothelioma cells transduced with either the catalytically inactive C91A *BAP1* mutant (C91A) or wild-type *BAP1* (WT).
DOI: https://doi.org/10.7554/eLife.30224.026

• Transparent reporting form
DOI: https://doi.org/10.7554/eLife.30224.027

## Major datasets

The following previously published dataset was used:

| Author(s) | Year | Dataset title | Dataset URL | Database, license, and accessibility information |
|---|---|---|---|---|
| Garnett MJ, Edelman EJ, Heidorn SJ, Greenman CD, Dastur A, Lau KW | 2012 | Data from the Cell Lines Project, V83 | http://cancer.sanger.ac.uk/cell_lines/download | Available at the Catalogue of Somatic Mutations in Cancer on registration and login. Downloads by academic and non-profit organisations are free but for-profit organisations are required to pay a license fee. |

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
