## [Decision Letter]

Thank you for submitting your article "Loss of functional BAP1 is a biomarker for TRAIL sensitivity in cancer" for consideration by *eLife*. Your article has been reviewed by three peer reviewers, and the evaluation has been overseen by a Reviewing Editor and Charles Sawyers as the Senior Editor. The following individuals involved in review of your submission have agreed to reveal their identity: Andrew Thorburn (Reviewer #1); Dan Longley (Reviewer #3).

The reviewers have discussed the reviews with one another and the Reviewing Editor has drafted this decision to help you prepare a revised submission.

Summary:

Kolluri et.al. present convincing data arising from a drug screen in a panel of mesothelioma cell lines that the nuclear deubiquitinase BAP1 regulates TRAIL sensitivity in mesothelioma cells (and some other tumor types too.) Loss of BAP1 expression, which occurs in some cancers, could potentially serve as a biomarker for sensitivity to agonists that target TRAIL receptors (TRAIL R). As noted by the authors TRAIL R-targeted therapeutics have been quite widely tested and usually shown to be safe, however the lack of an ability to identify patients who are likely to benefit from these treatments has stymied development of this class of drugs. Therefore, the area that the authors are addressing is potentially quite important. Reviewers found the data presented to be quite convincing and to support the authors' case that BAP1 can affect TRAIL sensitivity most likely through epigenetic regulation of gene expression by a mechanism involving the AXLS1 protein. However, a number of concerns were raised about the ultimate power of BAP1 status to serve as a biomarker in the clinic, as well as the minimal mechanistic insight provided in the report.

After thorough discussion, the reviewers agreed to invite a resubmission of a revised manuscript addressing the following major points:

1) A main concern with the paper in its current form is that it doesn't represent a major advance for the field, in the sense that there are many genes and processes that have been shown to confer TRAIL sensitivity and proposed as potential biomarkers, yet none of them so far have worked out to be useful. Reviewers agreed that more evidence is needed in this regard. First, it's not clear whether the effects are really specific to TRAIL. Many perturbations that make tumor cells more sensitive to apoptosis will appear to confer sensitivity on a canonical apoptosis inducer like TRAIL. Therefore, reviewers request a series of experiments addressing this issue, to define whether or not BAP1 mutation/loss is a general sensitizer to death ligands and/or general apoptotic stimuli. Reviewers suggest testing a different death ligand, such as a Fas agonist (e.g. CH-11 antibody) or, alternatively, a TNF/IAP antagonist combination. For the general apoptotic stimulus, the reviewers propose disease-relevant genotoxic agents such as pemetrexed or cisplatin.

2) Second, reviewers agreed that more work should be performed to establish the notion of BAP1 status as a biomarker for TRAIL efficacy. Although the pattern presented is very strong, many BAP1 positive cells are sensitive to TRAIL and vice versa, indicating that other factors in addition to BAP1 define sensitivity to TRAIL. The report will be significantly strengthened by performing cause-effect relationship experiments in a much larger number of cell lines. More specifically, reviewers would like to see the BAP1 depletion experiments done in a larger panel of cell lines. The report currently shows the effect of BAP1 knockdown in only two cell lines: H226 and MDA-MB-231. All other cause-effect experiments involve artifact-prone overexpression experiments of BAP1 (wt or various mutants). Given the ready access to cell lines and proven shRNAs for BAP1, the conclusions would be significantly strengthened by testing the effect of BAP1 depletion in two additional MM cell lines that are BAP1 positive and TRAIL-resistant. Additionally, BAP1 mutations are most frequent among kidney clear cell carcinomas (CRCC) estimated at 10% by tumorportal.org. The impact of the report would be significantly increased by measuring TRAIL sensitivity upon BAP1 depletion in two CCRC cell lines expressing wild type BAP1.

3) Third, the clinical relevance would be increased by testing a large number of patient samples in the ex vivo analyses in Figure 3 and/or to use a more clinically relevant TRAIL receptor agonist.

4) Another major issue was the minimal mechanistic insight as to how BAP1 status regulates sensitivity to TRAIL. What is the impact of BAP1 on known regulators and effectors of the TRAIL receptor pathway? Some data and discussion on the relationship between BAP1 mutation and gene expression of TRAIL-R1/2, FADD, FLIP, caspase-8, etc. is needed. What happens to these genes when WT versus mymutant BAP1 is overexpressed in BAP1 negative cell line?

5) Additionally, insight into how the ASXL1 connection translates into increased cell death should be provided, such as additional measures of key cell death signaling nodes as described in #4 above in experiments where ASXL1 activity/expression has been manipulated.

6) In the absence of true validation of BAP1 status as a biomarker in the clinical setting, the title was deemed overambitious, and reviewers request a more accurate title and some attenuation of some of the conclusions describing BAP1 as a biomarker.

---

## [Author Response]

[…] After thorough discussion, the reviewers agreed to invite a resubmission of a revised manuscript addressing the following major points:1) A main concern with the paper in its current form is that it doesn't represent a major advance for the field, in the sense that there are many genes and processes that have been shown to confer TRAIL sensitivity and proposed as potential biomarkers, yet none of them so far have worked out to be useful. Reviewers agreed that more evidence is needed in this regard. First, it's not clear whether the effects are really specific to TRAIL. Many perturbations that make tumor cells more sensitive to apoptosis will appear to confer sensitivity on a canonical apoptosis inducer like TRAIL. Therefore, reviewers request a series of experiments addressing this issue, to define whether or not BAP1 mutation/loss is a general sensitizer to death ligands and/or general apoptotic stimuli. Reviewers suggest testing a different death ligand, such as a Fas agonist (e.g. CH-11 antibody) or, alternatively, a TNF/IAP antagonist combination. For the general apoptotic stimulus, the reviewers propose disease-relevant genotoxic agents such as pemetrexed or cisplatin.

We agree and highlight in our manuscript that to date a key limiting factor of TRAIL therapeutics has been the lack of a validated biomarker for sensitivity. However, no approach thus far has included the search for a sensitising mutation using an unbiased drug screen in sequenced cell lines. Importantly the retrospective identification of drug sensitising mutations has re-directed the use of other anticancer therapies with initially disappointing results in unselected populations.

To demonstrate the specificity of *BAP1* as a sensitising mutation to TRAIL, and other death receptor agonists, we have treated a panel of *BAP1* mutant and wild-type MM lines with additional apoptotic stimuli as suggested. No sensitising association with *BAP1* was observed for pemetrexed or cisplatin, current first line agents for the treatment of MM (Figure 1—figure supplement 2). A marginal trend towards increased sensitivity in *BAP1* mutant MM lines in response to treatment with the agonistic FAS receptor antibody CH11 and a TNF-α/IAP inhibitor combination was observed. This was not however as pronounced as that observed with rTRAIL or the multivalent death receptor 5 superagonist MEDI3039 (Figure 1—figure supplement 2). Thus, while the significant sensitising association observed in the screen appears most specific to death receptor agonists, the trend observed with other TNF superfamily agonists indicates the BAP1-rTRAIL association to be mediated by an underlying mechanism common to this family, such as the cytoplasmic extrinsic apoptotic machinery.

The following text was added to the Results section:

“No sensitising association with *BAP1* was observed for pemetrexed or cisplatin, current first line agents for the treatment of MM (Figure 1—figure supplement 2). […] Thus, while the significant sensitising association observed in the screen appears most specific to death receptor agonists, the trend observed with other TNF superfamily agonists indicates the BAP1-rTRAIL association to be mediated by an underlying mechanism common to this family, such as the cytoplasmic extrinsic apoptotic machinery.”

2) Second, reviewers agreed that more work should be performed to establish the notion of BAP1 status as a biomarker for TRAIL efficacy. Although the pattern presented is very strong, many BAP1 positive cells are sensitive to TRAIL and vice versa, indicating that other factors in addition to BAP1 define sensitivity to TRAIL. The report will be significantly strengthened by performing cause-effect relationship experiments in a much larger number of cell lines. More specifically, reviewers would like to see the BAP1 depletion experiments done in a larger panel of cell lines. The report currently shows the effect of BAP1 knockdown in only two cell lines: H226 and MDA-MB-231. All other cause-effect experiments involve artifact-prone overexpression experiments of BAP1 (wt or various mutants). Given the ready access to cell lines and proven shRNAs for BAP1, the conclusions would be significantly strengthened by testing the effect of BAP1 depletion in two additional MM cell lines that are BAP1 positive and TRAIL-resistant. Additionally, BAP1 mutations are most frequent among kidney clear cell carcinomas (CRCC) estimated at 10% by tumorportal.org. The impact of the report would be significantly increased by measuring TRAIL sensitivity upon BAP1 depletion in two CCRC cell lines expressing wild type BAP1.

We agree that factors other than *BAP1* can affect TRAIL sensitivity as evidenced in our panel of MM lines (Figure 1). This is not unusual, noting that TKI inhibition of EGFR activating mutant lung cancers show response rates of around 70%. We have strengthened our data by extending the knockdown of *BAP1* wild-type cell lines to include four MM lines, two clear cell renal carcinoma lines and one breast cancer line.

We have amended the following text in the Results section:

“To determine if knockdown of *BAP1* in wild-type MM cells led to TRAIL sensitivity, we silenced *BAP1* expression in fourwt *BAP1* MM cell lines using a lentiviral shRNA construct. […] Notably, knockdown of BAP1 in two CCRC lines resulted in increased sensitivity to rTRAIL in addition to the MDAMB-231 breast cancer line (Figure 2 and Figure 2—figure supplement 2 and Figure 2—figure supplement 3).”

3) Third, the clinical relevance would be increased by testing a large number of patient samples in the ex vivo analyses in Figure 3 and/or to use a more clinically relevant TRAIL receptor agonist.

Since review we have obtained 7 video assisted thoracoscopic surgery (VATS) pleural biopsy samples from treatment naïve patients with a suspected diagnosis of MM. Four of these proved to comprise benign tissue only. We attempted to assess apoptosis in response to MEDI3039 treatment in explants from the remaining three. However, as these were VATS rather than pleurectomy biopsies, the tissue volume was extremely small precluding appropriate immunohistochemical analysis. Successful explant data is ultimately subject to the frequency of pleurectomy cases performed at our local surgical centre and within the suggested time period for resubmission there were none. We agree however that an expanded cohort would strengthen the clinical relevance and continue with our explant programme to this end.

4) Another major issue was the minimal mechanistic insight as to how BAP1 status regulates sensitivity to TRAIL. What is the impact of BAP1 on known regulators and effectors of the TRAIL receptor pathway? Some data and discussion on the relationship between BAP1 mutation and gene expression of TRAIL-R1/2, FADD, FLIP, caspase-8, etc. is needed. What happens to these genes when WT versus mymutant BAP1 is overexpressed in BAP1 negative cell line?

We have extended our mechanistic data as suggested. We have compared gene expression data from a *BAP1* null MM line with overexpression of *BAP1* wild-type or the catalytically inactive *BAP1* C91A. Signaling pathway impact analysis highlighted apoptosis as being significantly altered by BAP1 function. Thus we proceeded as suggested to specifically assess the impact of BAP1 function on the death receptor pathway at a gene expression and protein level. This data supports a role for BAP1 in the modulation of transcription of death receptor pathway proteins.

We have added the following text to the Results section:

“We therefore compared differential gene expression data from *BAP1*-null H226 cells transduced with the C91A *BAP1* mutant or with wild-type *BAP1,* and carried out a signalling pathway impact analysis SPIA ((Figure 2—figure supplement 7 and 8 [SPIA_H226 C91A mutant vs. WT]) (http://www.genome.jp/dbget-bin/www_bget?path:map04210). […] Flow cytometry analysis confirmed reduced DR4 and DR5 expression in C91A BAP1 transduced relative to wild-type *BAP1* transduced cells. Knockdown of *BAP1* in the *BAP1* wild-type H2818 line resulted in a significant increase in DR4 expression only (Figure 2).”

5) Additionally, insight into how the ASXL1 connection translates into increased cell death should be provided, such as additional measures of key cell death signaling nodes as described in #4 above in experiments where ASXL1 activity/expression has been manipulated.

We have conducted immunoblot analysis of the death receptor pathway on *BAP1* null MM cells transduced with the ASXL1/2 binding mutant ΔASXL *BAP1.* The changes observed in this analysis are consistent with that of those cells transduced with the catalytically inactive C91A *BAP1* mutant. This supports the role of both functions, deubiquitinase activity and ASXL binding, in the regulation of expression of components of the death receptor pathway.

These changes to the manuscript are detailed above in point 4.

6) In the absence of true validation of BAP1 status as a biomarker in the clinical setting, the title was deemed overambitious, and reviewers request a more accurate title and some attenuation of some of the conclusions describing BAP1 as a biomarker.

We have amended the title to:

“Loss of functional BAP1 augments TRAIL sensitivity in cancer cells.”

We have minimised the reference of BAP1 as a biomarker replacing such instances with reference to its use as a ‘genomic stratification tool’ or ‘potential biomarker’.